# Synthesis and Biological Evaluation of Novel Amino and Amido Substituted Pentacyclic Benzimidazole Derivatives as Antiproliferative Agents

**DOI:** 10.3390/ijms25042288

**Published:** 2024-02-14

**Authors:** Nataša Perin, Marjana Gulin, Marija Kos, Leentje Persoons, Dirk Daelemans, Ivana Fabijanić, Marijana Radić Stojković, Marijana Hranjec

**Affiliations:** 1Department of Organic Chemistry, Faculty of Chemical Engineering and Technology, University of Zagreb, Marulićev trg 20, HR-10000 Zagreb, Croatia; nperin@fkit.unizg.hr (N.P.); mgulin@fkit.unizg.hr (M.G.); mkos@fkit.unizg.hr (M.K.); 2KU Leuven, Department of Microbiology and Immunology, Laboratory of Virology and Chemotherapy, Rega Institute, 3000 Leuven, Belgium; leentje.persoons@kuleuven.be (L.P.); dirk.daelemans@kuleuven.be (D.D.); 3Ruđer Bošković Institute, Division of Organic Chemistry and Biochemistry, Bijenička cesta 54, HR-10000 Zagreb, Croatia; ifabijanic@irb.hr (I.F.); mradic@irb.hr (M.R.S.)

**Keywords:** antiproliferative activity, amines, amides, benzimidazoles, interaction with DNA

## Abstract

Newly designed pentacyclic benzimidazole derivatives featuring amino or amido side chains were synthesized to assess their in vitro antiproliferative activity. Additionally, we investigated their direct interaction with nucleic acids, aiming to uncover potential mechanisms of biological action. These compounds were prepared using conventional organic synthesis methodologies alongside photochemical and microwave-assisted reactions. Upon synthesis, the newly derived compounds underwent in vitro testing for their antiproliferative effects on various human cancer cell lines. Notably, derivatives **6** and **9** exhibited significant antiproliferative activity within the submicromolar concentration range. The biological activity was strongly influenced by the N atom’s position on the quinoline moiety and the position and nature of the side chain on the pentacyclic skeleton. Findings from fluorescence, circular dichroism spectroscopy, and thermal melting assays pointed toward a mixed binding mode—comprising intercalation and the binding of aggregated compounds along the polynucleotide backbone—of these pentacyclic benzimidazoles with DNA and RNA.

## 1. Introduction

Nowadays, nitrogen heterocycles are the essential core in numerous synthetic and natural pharmacologically active compounds [1,2,3]. These compounds exhibit diverse biological activities, making them pivotal elements in constructing innovative therapeutic agents [4,5]. Recently, a growing focus has been on producing fused, benzannulated benzimidazole derivatives due to their demonstrated significance in natural, medicinal, and environmental sciences [6,7]. Cyclic benzimidazoles possess a highly conjugated, flat chromophore with outstanding spectroscopic properties, which are exploitable in medicinal chemistry [8]. Due to their structural resemblance to naturally occurring purines, benzimidazole derivatives possess the capability to readily interact with biomolecules such as DNA, RNA, or proteins within living systems [9]. The most commonly used classes of chemotherapeutic agents are those that directly interact with DNA [10]. Numerous critical intracellular processes, including transcription, regulation, and translation, rely on the interaction of small molecules with nucleic acid structures [11,12], thereby expanding the broad applications of small molecules in cancer therapy [13]. Aside from covalent bonding, various non-covalent reversible binding mechanisms exist, including intercalation, minor and major groove binding, and electrostatic interactions with the nucleic acid backbone [14].

Given that quinoline nuclei serve as crucial pharmacophores in medicinal chemistry, forming a structural component in a myriad of alkaloids and synthetic molecules with diverse biological functions [15,16,17], we have recently synthesized a series of compounds, including tetracyclic quinolone-fused. These include tetracyclic quinoline-fused benzimidazoles (Figure 1a) [18,19,20], tetracyclic imidazo[4,5-*b*]pyridine derivatives (Figure 1b) [21,22], and pentacyclic benzo[*b*]thieno[2,3-*b*]pyrido-[1,2-*a*]benzimidazoles (Figure 1c) [23]. These compounds exhibited significant antiproliferative activity, indicating substantial potential for enhancing such frameworks to develop superior antitumor agents. Through comprehensive studies involving cytostatic assessment, analysis of DNA/RNA interactions, inhibition of topoisomerase I and II, and proteomic profiling, we confirmed the impact of the substituent type (amidino, amido, cyano) and its position on the tetracyclic and pentacyclic structures on their biological activity and mode of action.

Benzimidazo[1,2-*a*]quinolines and their heteroaromatic counterparts, featuring positively charged amidino substituents, exhibited notable selectivity toward colon carcinoma cells within submicromolar inhibitory concentration ranges, along with the ability to intercalate into double-stranded DNA or RNA [24,25]. Moreover, we synthesized 2-amino-, 5-amino-, and 2,5-diaminobenzimidazo[1,2-*a*]quinoline-6-carbonitriles, varying the lengths of secondary or tertiary amino chains linked to the tetracyclic framework, significantly influencing their anti-tumoral activity. Derivatives with *N,N*-dimethylaminopropyl substitutions displayed heightened antiproliferative activity, unequivocally establishing them as potent DNA binders and intercalative agents. The nature and length of the amino side chain significantly impacted cellular uptake and nuclear targeting [18,19,20,21,22,23]. Pentacyclic derivatives containing piperazinyl nuclei also demonstrated robust antiproliferative activity. Translocating amino side chains from position 3 to 6 marginally enhanced antitumoral activity. Conversely, for tetracyclic benzimidazoles, derivatives with amino side chains at position 2 exhibited superior activity [18,19,20].

Considering the significant biological promise of benzannulated benzimidazoles, we have devised and prepared novel pentacyclic derivatives to serve as potent and innovative antiproliferative agents. These compounds were modified with diverse amino and amido side chains situated at various positions within the skeleton. Furthermore, we explored the influence of the N atom position on the quinoline nuclei. Additionally, we investigated the DNA/RNA binding mechanisms of the most active compounds.

## 2. Results and Discussion

### 2.1. Chemistry

All target amino-substituted compounds **6**–**9** and **19**–**22**, as well as amido-substituted compounds at positions **25**–**26** and **29**–**30** in the pentacyclic benzimidazole derivatives, along with their corresponding intermediates, were synthesized using two outlined synthetic procedures in Figure 1 and Figure 2.

The synthesis of 7-amino substituted pentacyclic derivatives commenced from commercially available 2-chloro-3-quinolinecarbonyl chloride **1**. This compound underwent condensation with 2-cyanomethylbenzimidazole **2**, yielding the corresponding acyclic hydroxyl-substituted acrylonitrile derivative **3** with a good reaction yield of 76% [26]. Upon thermal cyclization with *t*-KOBu in DMF, the 7-oxo substituted pentacyclic derivative 4 was obtained with an 83% yield. Subsequently, derivative **4** was successfully transformed into the 7-chloro substituted derivative **5** (yield: 12%), serving as the primary precursor for synthesizing the 7-amino substituted target compounds **6**–**9**. Using an uncatalyzed microwave-assisted amination method with the respective amines in acetonitrile at 170 °C (800 W), the corresponding 7-amino-6-cyano substituted pentacyclic benzimidazoles **6**–**9** were prepared, with yields ranging from 83% to 32%. The successful substitution of chlorine with the amine side chain was confirmed through NMR spectra, revealing proton signals associated with amino groups, methine, methylene, and methyl groups within the aliphatic portion, observable in both the 1H (3.91–1.80 ppm) and 13C NMR spectra (54.5–19.6 ppm).

To synthesize the 11-amino substituted pentacyclic derivatives **19**–**22**, the synthetic pathway was initiated with commercially available 3-quinolinecarboxaldehyde **10**. This compound underwent condensation with 2-cyanomethylbenzimidazole **2**, resulting in the formation of corresponding acyclic acrylonitrile **13** with a moderate reaction yield of 76%. Through thermal cyclization in sulfolane at 300 °C, the primary product obtained was the 11-fluoro substituted pentacyclic derivative **16**, serving as the main precursor, synthesized with a good yield of 78%. Following the previously mentioned uncatalyzed microwave-assisted amination technique, the corresponding 11-amino substituted pentacyclic derivatives **19**–**22** were synthesized with low to moderate (8–67%) yields.

Additionally, two distinct synthetic methods were employed to prepare the target 6-amido and 7-amido substituted pentacyclic derivatives. These methods differed in the positioning of the nitrogen atom within the quinoline moiety in the pentacyclic skeleton. To synthesize the 6-amido substituted benzo[*g*]benzo[4,5]imidazo-[1,2-*a*][1,8]naphthyridines **25**–**26**, the previously mentioned synthesis commencing from chloro-substituted 3-quinolinecarboxaldehyde **11** was utilized to prepare acyclic **14** and the pentacyclic substituted 6-cyano substituted derivative **17**.

Through acidic hydrolysis using concentrated sulfuric acid, the corresponding carboxylic acid **23** was successfully obtained with a commendable reaction yield of 97%.

This carboxylic acid was then employed in the reaction with thionyl chloride to yield pentacyclic carbonyl chloride **24** [27]. Subsequent reactions with suitable amines in dichloromethane (DCM) resulted in the synthesis of target pentacyclic amides **25**–**26**, although achieved in relatively low reaction yields.

For the synthesis of the alternative pentacyclic regioisomers, specifically the 7-amido substituted benzo[*g*]benzo[4,5]imidazo[1,2-*a*][1,5] naphthyridines **29**–**30**, the initial synthetic step involved preparing unsubstituted acyclic acrylonitrile **15** from quinoline-3-carbaldehyde, yielding a good reaction yield of 82%. The cyano-substituted cyclic derivative **18** was synthesized via photochemical cyclization in ethanol under irradiation for 4 h, resulting in a 72% yield. Using the previously mentioned standard synthetic techniques, acidic hydrolysis, and reaction with thionyl chloride, pentacyclic carbonyl chloride **28** was obtained with a moderate yield. The structural confirmation of both pentacyclic carbonyl chlorides **24** and **28** relied solely on IR spectroscopy TCL chromatography. Following the reaction with appropriate amines in dimethylformamide (DMF), 7-amido substituted benzo[*g*]benzo[4,5]imidazo[1,2-*a*][1,5] naphthyridines **29**–**30** were obtained with moderate reaction yields. Due to the synthetic issues, the piperazinyl substituted compound was not synthesized, while compound **30** was obtained via the interaction of DMF with carbonyl chloride **28** upon microwave heating. Structural verification of all synthesized compounds was carried out using 1H and 13C NMR spectroscopy, along with mass spectrometry (MS), high-resolution mass spectrometry (HRMS), elemental analysis, and UV/Vis spectroscopy. The structural characterization demonstrated that the observed chemical shifts in the 1H and 13C NMR spectra, as well as the H–H coupling constants, were consistent with the proposed structures. The formation of pentacyclic structures was confirmed by the disappearance of the signal associated with the proton of the NH group on the benzimidazole nuclei. Both photochemical and thermal cyclization processes were monitored using UV/Vis spectroscopy.

### 2.2. Biological Activity

#### 2.2.1. Antiproliferative Activity In Vitro

The antiproliferative potential of the pentacyclic benzimidazole derivatives, whether amino or amido substituted, was assessed in vitro against a diverse panel of cancer cell lines, including pancreatic adenocarcinoma (Capan-1), chronic myeloid leukemia (Hap-1), colorectal carcinoma (HCT-116), lung carcinoma (NCI-H460), acute lymphoblastic leukemia (DND-41), acute myeloid leukemia (HL-60), chronic myeloid leukemia (K-562), multiple myeloma (MM1.S), and non-Hodgkin lymphoma (Z-138). The resulting data are presented in Table 1 as IC_50_ values (50% inhibitory concentration), with the reference compound etoposide, a well-known chemotherapeutic and topoisomerase II inhibitor, included for comparison. When compared to the reference molecule, certain derivatives exhibited superior activity against specific cancer cell lines. In Table 1, several compounds displaying notable antiproliferative effects with inhibitory concentrations in the submicromolar range are highlighted. While most derivatives showed moderate to good activity, three compounds were devoid of antitumoral effects, even at the highest tested concentration (100 µM).

Derivative **6**, characterized by an amino side chain at position 7 of the pentacyclic skeleton with *N,N*-dimethylaminopropyl substitution, displayed the most potent activity. It demonstrated substantial inhibitory effects at submicromolar concentrations (IC_50_ 0.3–1.8 µM) across multiple cancer cell lines, designating it as a promising lead compound. Further refinement of its structure is warranted to optimize both activity and selectivity. Additionally, a structural analogue of derivative **6**, namely, compound **19**, featuring an amino side chain at position 11 of the pentacyclic skeleton with *N,N*-dimethylaminopropyl substitution, exhibited similar activity against K-562 and Z-138 cancer cells, also within the submicromolar range of inhibitory concentrations (IC_50_ 0.4 µM and 0.6 µM, respectively).

Furthermore, the pentacyclic derivative **9**, substituted with 7-piperazinyl, exhibited an overall IC_50_ range of 1.5–5.6 µM across the panel of cancer cell lines. Similarly, its structural analogue **22**, with a piperazinyl nucleus at position 11 of the pentacyclic skeleton, demonstrated IC_50_ values ranging from 1.2 to 8.4 µM.

When comparing the results for these structural analogues, featuring amino side chains at either position 7 or position 11 of the pentacyclic skeleton, a slight improvement in antiproliferative activity was observed when the amino side chain occupied position 7. Conversely, compounds such as the 7-piperidinyl substituted derivative **8** and the 11-fluoro and 11-unsubstituted cyano-substituted pentacyclic precursors **16** and **17** completely lacked antiproliferative activity across the panel of cancer cell lines (overall IC_50_ > 100 µM).

Within the amido-substituted compounds, the carboxamide derivative **25**, with an *N,N*-dimethylaminopropyl side chain, displayed the most pronounced activity, with an IC_50_ range spanning from 2.1 to 28.5 µM. This compound exhibited increased antiproliferative efficacy against the Z-138 cancer cell line (IC_50_ value of 2.1 µM). Upon comparing the outcomes for the two categories of amido-substituted regioisomers, it was noted that the 6-amido-benzo[g]benzo[4,5]imidazo [1,2-*a*][1,8]naphthyridines exhibited marginally superior activity compared to the 7-amido-benzo[*g*]benzo[4,5]imidazo[1,2-*a*][1,5]naphthyridines.

In summary, it is confirmed that the antiproliferative activity was markedly increased when the *N,N*-dimethylaminopropyl amino side chain was positioned at C-7 of the pentacyclic skeleton. Additionally, analogues with an amino side chain at C-11 of the cyano-substituted precursors, as well as 6-amido/7-amido pentacyclic derivatives, display overall enhanced activity (see Figure 2). Thus, both the position (C-7 or C-11) and the nature of the substituent (amino or amido) strongly influence the observed antiproliferative activity.

#### 2.2.2. Binding with Nucleic Acids

Four compounds, namely, **6**, **9**, **19**, and **22**, were selected based on their performance in the antitumoral evaluation for further investigation concerning their interaction with nucleic acids. These compounds were dissolved in DMSO at a concentration of c = 2 × 10^−3^ mol dm^−3^. The absorbance measurements of their buffered aqueous solutions indicated a direct proportionality to their concentrations up to c = 2 × 10^−5^ mol dm^−3^. This observation suggested that within this concentration range, the compounds did not form assemblies through intermolecular stacking interactions.

The absorption maxima and corresponding molar extinction coefficients (ε) were determined and are provided in Figure 3 and Appendix A). Fluorimetric measurements were conducted in a spectral region devoid of emission and excitation spectral overlap (Appendix A).

The investigation into the interaction of these compounds involved the use of calf thymus (ct)DNA, representing regular B-form DNA (with 58% AT base pairs and 42% GC base pairs), and AU homopolynucleotide (rArU), serving as a model for A-helical structure (RNA) [28,29,30]. Thermal melting (Tm) experiments provided crucial parameters, including the ΔTm value, representing the disparity between the Tm value of the free polynucleotide and the Tm of the complex formed with a small molecule [31].

Groove binding interactions can result in either substantial (positive ΔTm values) or minor stabilization or destabilization of DNA (negative ΔTm values). Conversely, positive ΔTm values are typically characteristic of intercalating small molecules. Upon evaluation, compounds **6**, **9**, **19**, and **22** exhibited moderate stabilizing effects on both DNA (ctDNA) and RNA (rArU), as illustrated in Table 2, Figure 4, and Appendix A. Among these compounds, compound **22**, which features a piperazine ring at C-11, demonstrated the most pronounced stabilizing effect on RNA.

Titrations with ctDNA and rArU resulted in a fluorescence decrease in **6**, **9**, **19,** and **22** (Figure 5; see Appendix A).

The binding constants (Ka) for the complexes formed between the ligands and DNA/RNA were determined from the fluorimetric titration data using the Scatchard equation, and the calculated values are detailed in Table 2. Notably, compounds **19** and **22**, both featuring an amino substituent (piperazine or dimethylaminopropylamino) at the C-11 position, displayed substantial binding affinities toward nucleic acids (both DNA and RNA). This increase in binding affinity was particularly pronounced for the AU homopolynucleotide (RNA).

Small fluorescence changes of derivatives **6** and **9** in the titration with ctDNA disabled and the accurate calculation of binding constants (see Appendix A).

Circular dichroism (CD) spectroscopy serves as a valuable tool in monitoring alterations in nucleic acid conformation following the introduction of small molecules. It also offers insights into the interaction mode based on the relative orientation of the small molecule and the chiral axis of the nucleic acid [34]. When interacting with nucleic acids, achiral small molecules like **6**, **9**, **19**, and **22** can induce a CD spectrum known as Induced Circular Dichroism (ICD). In the spectral region beyond 300 nm, where DNA exhibits no absorption (as depicted in Figure 6 and Appendix A), informative data regarding binding can be extracted. Upon the addition of compounds **6**, **9**, **19**, and **22**, a substantial reduction in CD intensity was observed in ctDNA at 275 nm and in RNA at 260 nm.

Compounds **6**, **9**, **19**, and **22** exhibited the emergence of negative Induced Circular Dichroism (ICD) bands, which were less prominent at lower ratios (r [compound]/[nucleotide phosphate]) and became more pronounced at higher ratios (r > 0.2), as depicted in Figure 6 and Appendix A. The induced circular dichroism was more conspicuous for DNA compared to RNA.

Both the CD spectroscopy and thermal melting analysis collectively suggest that the anticipated mode of binding to nucleic acids at lower ratios (r [compound]/[nucleotide phosphate] < 0.2) involves intercalation. However, at higher ratios (r > 0.2), there appears to be a shift toward binding of aggregated compounds along the polynucleotide backbone [35].

#### 2.2.3. Effects of Pentacyclic Benzimidazole Derivatives on Cancer Cell Cycle Regulation

To further validate and explore the impact of the newly synthesized derivatives on cancer cell proliferation, we investigated the effect of derivatives **6** and **19** on cell cycle regulation. Our findings show that both derivatives dose-dependently increase the percentage of cells in the G2/M phase following a 24 h treatment with a concomitant reduction in cells in the G1 phase of the cell cycle. The same pattern can be seen for the reference drug doxorubicin (DOXO), an antineoplastic agent known to intercalate into nucleic acids. As shown in Figure 7, the effect on cell cycle regulation was somewhat more pronounced in the NCI-H460 cell line (B) when compared to the Capan-1 cell line (A).

Alongside the G2/M arrest, an increasing percentage of cells in the sub-G1 phase was measured after treatment with derivatives **6** and **19**. An increase in the percentage of cells in the sub-G1 phase in a cell cycle experiment is typically indicative of apoptotic cell death since the sub-G1 phase represents a population of cells with DNA content lower than the G1 phase, often resulting from DNA fragmentation during apoptosis.

#### 2.2.4. Effects of Pentacyclic Benzimidazole Derivatives on Caspase-3 and -7 Activity

As suggested by the increased percentage of cells in the sub-G1 phase following exposure to derivatives **6** and **19**, our objective was to examine the potential of these pentacyclic benzimidazole derivatives to induce apoptosis in various cancer cell types. Therefore, we investigated the dose-dependent effects of **6** and **19** on pancreatic adenocarcinoma and lung carcinoma apoptosis.

As depicted in Figure 8, we observed an increase in caspase-3 and -7 activity in both cell lines after 24 h of incubation with 10 µM of each derivative, while lower doses (1 and 0.1 µM) exhibited no discernible effect. Notably, derivative **6** exerted a more pronounced effect on the NCI-H460 cell line compared to derivative **19**, whereas the opposite was true for the Capan-1 cell line. The reference drug doxorubicin (DOXO) was able to induce apoptosis, even at a concentration of 1 µM, and this was seen in both cancer cell lines.

## 3. Materials and Methods

### 3.1. Chemistry—General Methods

Chemicals and solvents used in this study were procured from commercial suppliers such as Aldrich and Acros. Melting points were determined using the SMP11 Bibby (Stockholm, Sweden) and Büchi 535 apparatus (Flawil, Switzerland). NMR spectra were acquired in DMSO-d6 solutions, utilizing TMS as an internal standard. The 1H and 13C NMR spectra were recorded on Varian Gemini 300 (Varian Medical Systems, Palo Alto, CA, USA) or Varian Gemini 600 (Varian Medical Systems, Palo Alto, CA, USA) spectrometers, operating at 300, 600, 150, and 75 MHz, respectively. Chemical shifts were reported in ppm (δ) relative to TMS. To ensure the purity of compounds, TLC analysis was routinely conducted using Merck silica gel 60F-254 glass plates. In preparative photochemical experiments, irradiation was conducted at room temperature utilizing a water-cooled immersion well with an “Origin Hanau” 400 W high-pressure mercury arc lamp, employing Pyrex glass as a filter. High-resolution mass spectrometry (HRMS) analysis was performed using a mass spectrometer (MALDI TOF/TOF 4800 plus analyzer, Applied Biosystems, Mundelein, IL, USA). Elemental analysis for carbon, hydrogen, and nitrogen was carried out using a Perkin-Elmer 2400 (PerkinElmer, Waltham, MA, USA) elemental analyzer. In cases where elemental analyses are represented by elemental symbols, the obtained analytical results were within 0.4% of the theoretical values.

### 3.2. Synthesis of 7-Amino Substituted Benzo[g]benzo[4,5]imidazo[1,2-a][1,8]-naphthyridine-6-carbonitriles

#### 3.2.1. (*E*)-2-benzimidazolyl-1-(2-chloroquinolin-3-yl)-2-isocyanoethen-1-ol **3**

A solution of 0.40 g (2.34 mmol) of 2-cyanomethylbenzimidazole **2** and 0.60 g (2.45 mmol) of 2-chloroquinoline-3-carbonyl chloride **1** in pyridine (6 mL) was refluxed for 1.5 h. The cooled mixture was poured into water (50 mL), and the resulting product was filtered off and recrystallized from ethanol to obtain a brown powder (1.24 g, 76%). m.p. > 300 °C. 1H NMR (DMSO-d6, 300 MHz): δ/ppm = 13.38 (s, 1H, OH), 12.62 (s, 1H, NHbenzim.), 9.30 (s, 1H, Harom.), 9.21–9.19 (m, 1H, Harom.), 8.34 (d, 1H, J = 8.61 Hz, Harom.), 8.30 (d, 1H, J = 8.52 Hz, Harom.), 8.00 (t, 1H, J = 7.65 Hz, Harom.), 7.72 (t, 1H, J = 7.12 Hz, Harom.), 7.58–7.55 (m, 1H, Harom.), 7.50–7.74 (m, 2H, Harom.); 13C NMR (DMSO-d6, 75 MHz): δ/ppm = not enough soluble; found: C, 65.60; H, 3.05; N, 16.36. Calc. for C19H11ClN4O: C, 65.81; H, 3.20; N, 16.16%.

#### 3.2.2. 7-Oxo-5,7-dihydrobenzo[g]benzo[4,5]imidazo[1,2-a][1,8]naphthyridine-6-carbo-nitrile **4**

A solution of 1.24 g (1.69 mmol) **3** and 1.20 g (11.00 mmol) *t*-KOBu in DMF (7 mL) was refluxed for 2 h. After cooling, the reaction mixture was evaporated under vacuum and dissolved in water (50 mL). Resulting product was filtered off and recrystallized from ethanol to obtain a light brown powder (0.93 g, 83%). m.p. >300 °C. ^1^H NMR (DMSO-*d*_6_, 300 MHz): δ/ppm = 13.31 (bs, 1H, NH), 9.35 (s, 1H, H_arom._), 8.47 (d, 1H, *J* = 7.92 Hz, H_arom._), 8.22 (d, 1H, *J* = 7.91 Hz, H_arom._), 8.01 (d, 1H, *J* = 8.53 Hz, H_arom._), 7.88 (dt, 1H, *J*_1_ = 8.52 Hz, *J_2_* = 1.42 Hz, H_arom._), 7.57–7.48 (m, 3H, H_arom._), 7.38–7.33 (m, 1H, H_arom._); ^13^C NMR (DMSO-*d*_6_, 100 MHz): δ/ppm = not enough soluble; found: C, 73.69; H, 3.15; N, 18.00. Calc. for C_19_H_10_N_4_O: C, 73.54; H, 3.25; N, 18.06%.

#### 3.2.3. 7-Chlorobenzo[g]benzo[4,5]imidazo[1,2-a][1,8]naphthyridine-6-carbonitrile **5**

A solution of 0.93 g (2.98 mmol) 7-oxo-5,7-dihydrobenzo[*g*]benzo[4,5]imidazo[1,2-*a*] [1,8]naphthyridine-6-carbonitrile **4** and 0.35 g (1.70 mmol) PCl_5_ in POCl_3_ (17 mL) was refluxed for 2 h. After cooling, the reaction mixture was evaporated under vacuum and dissolved in water (10 mL). Resulting product was filtered off and washed with water to obtain a yellow powder (0.12 g, 12%). m.p. 293–295 °C. ^1^H NMR (DMSO-*d*_6_, 300 MHz): δ/ppm = 9.46 (s, 1H, H_arom._), 9.29 (bs, 1H, H_arom._), 8.45 (d, 1H, *J* = 8.62 Hz, H_arom._), 8.37 (d, 1H, *J* = 7.61 Hz, H_arom._), 8.11 (t, 1H, *J* = 7.43 Hz, H_arom._), 8.04 (d, 1H, *J* = 7.10 Hz, H_arom._), 7.82 (t, 1H, *J* = 7.33 Hz, H_arom._), 7.71–7.60 (m, 2H, H_arom._); ^13^C NMR (DMSO-*d*_6_, 100 MHz): δ/ppm = 147.8, 145.0, 144.5, 143.9, 143.3, 139.5, 134.7, 131.3, 130.4, 128.2, 127.6, 126.2, 125.8, 125.4, 120.5, 117.3, 115.8, 113.8, 103.7; found: C, 69.26; H, 2.68; N, 17.29. Calc. for C_19_H_9_ClN_4_: C, 69.42; H, 2.76; N, 17.04%; MS: *m/z* = 351.0410 ([M + Na]^+^).

#### 3.2.4. General Method for Preparation of Compounds **6**–**9**

Compounds **6**–**9** were prepared using microwave irradiation at optimized reaction time with 800 W of power and 40 bar of pressure from compound 5 in acetonitrile (10 mL) with an excess of the corresponding amine added. After cooling, the reaction mixture was filtered off, and resulting product was separated by column chromatography on SiO_2_ using dichloromethane/methanol as eluent.

##### 7-((3-*N,N*-(dimethylamino)propyl)amino)benzo[*g*]benzo[4,5]imidazo[1,2-*a*][1,8]-naphthyridi- ne-6-carbonitrile **6**

Compound **6** was prepared using the described method from **5** (0.06 g, 0.20 mmol) and *N,N*-dimethylamino-propyl-1-amine (0.14 mL, 1.44 mmol) after 4 h of irradiation to yield 0.05 g (65%) of yellow powder. m.p. 250–253 °C. ^1^H NMR (DMSO-*d*_6_, 300 MHz): δ/ppm = 9.11 (s, 1H, H_arom._), 8.92 (d, 1H, *J* = 7.14 Hz, H_arom._), 8.67 (bs, 1H, NH), 8.11 (d, 1H, *J* = 8.40 Hz, H_arom._), 7.99–7.93 (m, 1H, *J* = 8.10 Hz, H_arom._), 7.93 (t, 1H, *J* = 8.16 Hz, H_arom._), 7.69–7.63 (m, 2H, H_arom._), 7.41–7.30 (m, 2H, H_arom._), 3.91 (t, 2H, *J* = 6.36 Hz, CH_2_), 2.50–2.46 (m, 2H, CH_2_), 2.25 (s, 6H, CH_3_), 1.98–1.89 (m, 2H, CH_2_); ^13^C NMR (DMSO-*d*_6_, 75 MHz): δ/ppm = 150.4, 147.2, 144.7, 135.3, 133.5, 131.6, 129.5, 127.9, 127.1, 124.9, 124.8, 122.6, 118.5, 117.8, 116.4, 113.2, 57.2, 45.8 (2C), 44.0, 27.0.

Found: C, 73.27; H, 5.50; N, 21.23. Calc. for C_24_H_22_N_6_: C, 73.07; H, 5.62; N, 21.30%; MS: *m/z* = 395.1998 ([M + H]^+^).

##### 7-(N-isobutylamino)benzo[g]benzo[4,5]imidazo [1,2-a][1,8]naphthyridine-6-carbo- nitrile **7**

Compound **7** was prepared using above-described method from **5** (0.06 g, 0.20 mmol) and isobutylamine (0.12 mL, 1.23 mmol) after 4 h of irradiation to yield 0.05 g (72%) of yellow powder. m.p. 296–297 °C. ^1^H NMR (DMSO-*d*_6_, 300 MHz): δ/ppm = 9.51 (s, 1H, H_arom._), 9.01 (dd, 1H, *J*_1_ = 6.71 Hz, *J*_1_ = 2.24 Hz, H_arom._), 8.42 (t, 1H, *J* = 6.22 Hz, NH_amin_), 8.20 (d, 1H, *J* = 8.42 Hz, H_arom._), 8.07 (d, 1H, *J* = 7.82 Hz, H_arom._), 8.00–7.94 (m, 1H, H_arom._), 7.72–7.67 (m, 2H, H_arom._), 7.42–7.34 (m, 2H, H_arom._), 3.70 (t, 2H, *J* = 6.62 Hz, CH_2_), 2.29–2.13 (m, 1H, CH), 1.05 (d, 6H, *J* = 6.62 Hz, CH_3_); ^13^C NMR (DMSO-*d*_6_, 75 MHz): δ/ppm = 150.0, 149.1, 146.9, 144.4, 144.3, 135.1, 132.9, 131.2, 128.9, 127.4, 126.5, 124.6, 124.3, 122.1, 118.0, 117.1, 115.9, 112.7, 51.5, 28.3, 19.6 (2C); found: C, 75.39; H, 5.40; N, 19.21. Calc. for C_23_H_19_N_4_: C, 75.59; H, 5.24; N, 19.16%; MS: *m/z* = 366.1703 ([M + H]^+^).

##### 7-(Piperidin-1-yl)benzo[g]benzo[4,5]imidazo[1,2-a][1,8]naphthyridine-6-carbo- Nitrile **8**

Compound **8** was prepared using above-described method from **5** (0.06 g, 0.20 mmol) and piperidine (0.13 mL, 1.27 mmol) after 4 h of irradiation to yield 0.06 g (87%) of yellow powder. m.p. > 300 °C. ^1^H NMR (DMSO-*d*_6,_ 400 MHz): δ/ppm = 9.19 (dd, *J_1_* = 6.42 Hz, *J_2_* = 2.93 Hz, 1H, H_arom._), 9.05 (s, 1H, H_arom._), 8.36 (d, *J* = 8.11 Hz, 1H, H_arom._), 8.29 (d, *J* = 8.43 Hz, 1H, H_arom._), 8.03 (t, *J* = 8.32 Hz, 1H, H_arom._), 7.86 (dd, *J_1_* = 6.32 Hz, *J_2_* = 2.83 Hz, 1H, H_arom._), 7.74 (t, *J* = 7.20 Hz, 1H, H_arom._), 7.54–7.49 (m, 2H, H_arom._), 3.72 (t, *J* = 5.21 Hz, 4H, CH_2_), 1.93 (bs, 4H, CH_2_), 1.80 (bs, 2H, CH_2_); ^13^C NMR (DMSO-*d*_6,_ 100 MHz): δ/ppm = 158.5, 147.9, 147.4, 145.7, 144.2, 138.7, 133.7, 131.3, 130.4, 127.9, 126.9, 123.8, 119.3, 116.9, 116.4, 115.7, 54.4 (2C), 26.4 (2C), 24.1; found: C, 76.20; H, 5.15; N, 18.65. Calc. for C_24_H_19_N_5_: C, 76.37; H, 5.07; N, 18.55%; MS: *m/z* = 400.1519 ([M + Na]^+^).

##### 7-(Piperazin-1-yl)benzo[g]benzo[4,5]imidazo[1,2-a][1,8]naphthyridine-6-carbo- Nitrile **9**

Compound **9** was prepared using above-described method from **5** (0.06 g, 0.20 mmol) and piperazine (0.11 g, 1.27 mmol) after 4 h of irradiation to yield 0.02 g (32%) of yellow powder. m.p. 274–277 °C. ^1^H NMR (DMSO-*d*_6_, 300 MHz): δ/ppm = 9.09–9.07 (m, 1H, H_arom._), 8.98 (s, 1H, H_arom._), 8.28 (d, 1H, *J* = 8.31 Hz, H_arom._), 8.18 (d, 1H, *J* = 8.13 Hz, H_arom._), 7.97 (t, 1H, *J* = 7.38 Hz, H_arom._), 7.83–7.80 (m, 1H, H_arom._), 7.68 (t, 1H, *J* = 7.42 Hz, H_arom._), 7.47 (bs, 2H, H_arom._), 3.64 (s, 4H, CH_2_), 3.08 (s, 4H, CH_2_); ^13^C NMR (DMSO-*d*_6_, 75 MHz): δ/ppm = 157.9, 147.9, 147.3, 145.5, 144.4, 138.6, 133.6, 131.4, 130.3, 127.8, 126.8, 125.3, 125.1, 123.7, 119.4, 116.8, 116.4, 115.3, 54.5 (2C), 46.7 (2C); found: C, 73.10; H, 4.88; N, 22.00. Calc. for C_23_H_18_N_6_: C, 73.00; H, 4.79; N, 22.21%; MS: *m/z* = 401.1507 ([M + Na]^+^).

### 3.3. Synthesis of 11-Amino Substituted Benzo[g]benzo[4,5]imidazo[1,2-a][1,8]-naphthyridine-6-carbonitriles

#### 3.3.1. (*E*)-2-(1H-benzo[d]imidazol-2-yl)-3-(2-chloro-7-fluoroquinolin-3-yl)acrylonitrile **13**

Compound **13** was prepared from **2** (0.38 g, 2.39 mmol) and 2-chloro- 7-fluoroquinoline-3-carbaldehyde **10** (0.50 g, 2.39 mmol) and few drops of piperidine in absolute ethanol (7 mL) after refluxing for 2 h and recrystallization from ethanol to yield 0.43 (52%) of yellow powder. m.p. > 300 °C. ^1^H NMR (DMSO-*d*_6_, 300 MHz): δ/ppm = 13.43 (s, 1H, NH_benz._), 9.18 (s, 1H, H_etenil_), 8.57 (s, 1H, H_arom._), 8.34 (dd, 1H, *J*_1_ = 9.06 Hz, *J*_2_ = 6.24 Hz, H_arom._), 7.87 (dd, 1H, *J*_1_ = 10.05 Hz, *J*_2_ = 2.37 Hz, H_arom._), 7.76–7.69 (m, 3H, H_arom._), 7.32–7.29 (m, 2H, H_arom._); ^13^C NMR (DMSO-*d*_6_, 75 MHz): δ/ppm = 164.6 (d, *J* = 250.7 Hz), 162.9, 150.4, 148.8, 148.6, 146.8, 140.3 (2C), 139.8 (2C), 132.3 (d, *J* = 10.5 Hz), 125.9, 124.2, 119.2 (d, *J* = 25.2 Hz), 115.5, 112.6 (d, *J* = 21.4 Hz), 108.7; found: C, 68.78; H, 3.40; N, 16.84. Calc. for C_19_H_11_ClN_4_: C, 68.99; H, 3.35; N, 16.94%.

#### 3.3.2. 11-Fluorobenzo[g]benzo[4,5]imidazo[1,2-a][1,8]naphthyridine-6-carbonitrile **16**

Compound **13** (0.50 g, 1.60 mmol) was dissolved in 3 mL of sulfolane, and reaction mixture was heated for 15 min at 280 °C. The cooled mixture was poured into water (15 mL), and the resulting product was filtered off and recrystallized from ethanol (150 mL) to obtain a yellow powder (0.57 g, 78%). m.p. > 300 °C. ^1^H NMR (DMSO-*d*_6_, 300 MHz): δ/ppm = 9.16–9.13 (m, 1H, H_arom._), 9.08 (s, 1H, H_arom._), 8.77 (s, 1H, H_arom._), 8.31 (dd, 1H, *J_1_* = 6.39 Hz, *J_2_* = 8.97 Hz, H_arom._), 7.98–7.95 (m, 2H, H_arom._), 7.66–7.56 (m, 3H, H_arom._); ^13^C NMR (DMSO-*d*_6_, 151 MHz): δ/ppm = 164.8 (d, *J* = 253.6 Hz), 148.5, 145.9, 145.0, 143.4, 140.4, 140.1, 132.2 (d, *J* = 10.6 Hz), 130.7, 125.4, 124.5, 122.7, 119.9, 117.3 (d, *J* = 25.8 Hz), 116.9, 115.7, 114.8, 111.8 (d, *J* = 21.4 Hz), 102.4; found: C, 73.27; H, 2.81; N, 17.78. Calc. for C_19_H_9_FN_4_: C, 73.07; H, 2.90; N, 17.94%; MS: *m/z* = 313.0882 ([M + H]^+^).

#### 3.3.3. General Method for Preparation of Compounds **19**–**22**

Compounds **19**–**22** were prepared using microwave irradiation at optimized reaction time with 800 W of power and 40 bar of pressure from compound 16 in acetonitrile (10 mL) with an excess of the corresponding amine added. After cooling, the reaction mixture was filtered off, and resulting product was separated by column chromatography on SiO_2_ using dichloromethane/methanol as eluent.

##### 11-((3-*N,N*-(dimethylamino)propyl)amino)benzo[g]benzo[4,5]imidazo[1,2-a][1,8]-naphthyridine-6-carbonitrile **19**

Compound **19** was prepared using above-described method from **16** (0.07 g, 0.22 mmol) and *N,N*-dimethylamino-propyl-1-amine (0.18 mL, 1.56 mmol) after 3 h of irradiation to yield 0.04 g (50%) of yellow powder. m.p. 230–235 °C. ^1^H NMR (DMSO-*d*_6_, 300 MHz): δ/ppm = 9.22–9.19 (m, 1H, H_arom._), 8.58 (s, 2H, H_arom._), 7.99–7.79 (m, 1H, H_arom._), 7.74 (d, 1H, *J* = 9.01 Hz, H_arom._), 7.58–7.54 (m, 2H, H_arom._), 7.25 (t, 1H, *J* = 4.72 Hz, NH_amin_), 7.06 (d, 1H, *J* = 7.22 Hz, H_arom._), 6.86 (s, 1H, H_arom._), 3.26 (q, 2H, *J* = 5.64 Hz, CH_2_), 2.38 (t, 2H, *J* = 6.81 Hz, CH_2_), 2.20 (s, 6H, CH_3_), 1.87–1.75 (m, 2H, CH_2_); ^13^C NMR (DMSO-*d*_6_, 151 MHz): δ/ppm = 153.5, 150.8, 145.9, 145.6, 143.5, 140.2, 138.9, 130.6, 129.9, 124.9, 123.4, 119.3, 118.8, 116.9, 115.8, 111.1, 100.7, 97.1, 56.7, 45.2 (2C), 40.7, 26.3; found: C, 73.17; H, 5.40; N, 21.45. Calc. for C_24_H_22_N_6_: C, 73.07; H, 5.62; N, 21.30%; MS: *m/z* = 395.1986 ([M + H]^+^).

##### 11-(N-isobutylamino)benzo[g]benzo[4,5]imidazo[1,2-a][1,8]naphthyridine-6-carbonitrile **20**

Compound **20** was prepared using above-described method from **16** (0.21 g, 0.66 mmol) and isobutylamine (0.468 mL, 4.71 mmol) after 3 h of irradiation to yield 0.12 g (54%) of yellow powder. m.p. 286–288 °C. ^1^H NMR (DMSO-*d*_6_, 300 MHz): δ/ppm = 9.26–9.21 (m, 1H, NH), 8.60 (s, 2H, H_arom_), 7.96–7.91 (m, 1H, H_arom._), 7.77 (d, 1H, *J* = 9.03 Hz, H_arom._), 7.59–7.52 (m, 2H, H_arom._), 7.26 (t, 1H, *J* = 5.32 Hz, H_arom._), 7.15 (dd, 1H, *J_1_* = 9.01 Hz, *J_2_* = 2.06 Hz, H_arom._), 6.89 (s, 1H, H_arom._), 3.09 (t, 2H, *J* = 6.09 Hz, CH_2_), 2.06–1.92 (m, 2H, CH_2_), 1.04 (d, 6H, *J* = 6.63 Hz, CH_3_); ^13^C NMR (DMSO-*d*_6_, 75 MHz): δ/ppm = 154.4, 151.2, 146.7, 144.0, 140.6, 139.4, 131.2, 130.5, 125.5, 123.9, 120.0, 119.8, 119.4, 117.5, 116.2, 111.7, 101.6, 97.8, 50.8, 27.9, 20.8 (2C); found: C, 75.36; H, 5.41; N, 19.22. Calc. for C_23_H_19_N_5_: C, 75.59; H, 5.24; N, 19.16%; MS: *m/z* = 366.1720 ([M + H]^+^).

##### 11-(Piperidin-1-yl)benzo[g]benzo[4,5]imidazo[1,2-a][1,8]naphthyridine-6-carbo-nitrile **21**

Compound **21** was prepared using above-described method from **16** (0.21 g, 0.66 mmol) and piperidine (0.47 mL, 4.68 mmol) after 3 h of irradiation to yield 0.062 g (8%) of yellow powder. m.p. > 300 °C.

^1^H NMR (DMSO-*d*_6_, 300 MHz): δ/ppm = 9.43–9.38 (m, 1H, H_arom._), 8.83 (s, 1H, H_arom._), 8.76 (s, 1H, H_arom._), 8.02 (d, 1H, *J* = 9.32 Hz, H_arom._), 7.99–7.96 (m, 1H, H_arom._), 7.64–7.57 (m, 3H, H_arom._), 7.48 (s, 1H, H_arom._), 3.65 (bs, 4H, CH_2_), 1.70 (bs, 6H, CH_2_); ^13^C NMR (DMSO-*d*_6_, 75 MHz): δ/ppm = 154.6, 150.8, 146.5, 144.1, 140.7, 139.7, 131.3, 130.8, 125.6, 124.2, 119.9, 119.5, 118.6, 117.7, 116.1, 112.8, 106.5, 98.7, 48.5 (2C), 25.6 (2C), 24.4; found: C, 76.20; H, 5.17; N, 18.63. Calc. for C_24_H_19_N_5_: C, 76.37; H, 5.07; N, 18.55%; MS: *m/z* = 378.1720 ([M + H]^+^).

##### 11-(Piperazin-1-yl)benzo[g]benzo[4,5]imidazo[1,2-a][1,8]naphthyridine-6-carbo-nitrile **22**

Compound **22** was prepared using above-described method from **16** (0.21 g, 0.66 mmol) and piperazine (0.30 g, 4.71 mmol) after 3 h of irradiation to yield 0.20 g (67%) of yellow powder. m.p. > 300 °C. ^1^H NMR (DMSO-*d*_6_, 300 MHz): δ/ppm = 9.24 (d, 1H, *J* = 7.08 Hz, H_arom._), 8.70 (s, 1H, H_arom._), 8.63 (s, 1H, H_arom._), 7.96–7.90 (m, 2H, H_arom._), 7.58–7.48 (m, 3H, H_arom._), 7.27 (s, 1H, H_arom._), 3.52 (bs, 4H, CH_2_), 2.96 (s, 4H, CH_2_), 2.86 (s, 1H, NH); ^13^C NMR (DMSO-*d*_6_, 75 MHz): δ/ppm = 154.7, 150.4, 146.4, 145.9, 143.9, 140.8, 139.8, 131.2, 130.7, 125.6, 124.2, 119.9, 119.6, 118.4, 117.6, 116.2, 112.9, 106.6, 98.8, 47.9, 45.6 (2C), 43.8; found: C, 73.13; H, 4.86; N, 22.01. Calc. for C_23_H_18_N_6_: C, 73.00; H, 4.79; N, 22.21%; MS: *m/z* = 379.1660 ([M + H]^+^).

### 3.4. Synthesis of Amido Substituted benzo[g]benzo[4,5]imidazo[1,2-a][1,8]naphthyridines and benzo[g]benzo[4,5]imidazo[1,2-a][1,5]naphthyridines

#### 3.4.1. General Method for the Synthesis of Compounds **14** and **15**

Solution of equimolar amounts of 2-cyanomethylbenzimidazole, corresponding aromatic aldehydes (**10** or **12**), and a few drops of piperidine in absolute ethanol was refluxed for 2 h. Following reaction, the mixture was cooled to room temperature, and the crude product was filtered off and recrystallized from ethanol.

##### (E)-2-(1H-benzo[d]imidazol-2-yl)-3-(2-chloroquinolin-3-yl)acrylonitrile **14**

Compound **14** was prepared using above-described method from 2-cyano- methylbenzimidazole **2** (0.25 g, 1.56 mmol) and 2-chloroquinoline- 3-carbaldehyde **11** (0.30 g, 1.56 mmol) in absolute ethanol (6 mL) to yield 0.45 g (86%) of orange powder. m.p. 294–297 °C. ^1^H NMR (600 MHz, DMSO): δ/ppm = 13.38 (s, 1H, NH_benzimid._), 9.12 (s, 1H, H_arom._), 8.56 (s, 1H, H_arom._), 8.19 (d, *J* = 7.92 Hz, 1H, H_arom._), 8.04 (d, *J* = 8.43 Hz, 1H, H_arom._), 7.94 (t, *J* = 7.82 Hz, 1H, H_arom._), 7.76 (t, *J* = 7.50 Hz, 1H, H_arom._), 7.75 (bs, 1H, H_arom._) 7.59 (bs, 1H, H_arom._), 7.31 (bs, 1H, H_arom._), 7.27 (bs, 1H, H_arom._); ^13^C NMR (150 MHz, DMSO): δ/ppm = 147.1, 145.2, 145.1, 143.2, 141.0, 140.4, 133.4, 130.8, 129.3, 128.0, 126.8, 125.4, 125.3, 124.6, 119.9, 116.8, 116.3, 115.0, 102.3; found: C, 69.05; H, 3.54; N, 16.70. Calc. for C_19_H_11_ClN_4_: C, 68.99; H, 3.35; N, 16.94%; MS: *m/z* = 331.0750 ([M + H]^+^).

##### (E)-2-(1H-benzo[d]imidazol-2-yl)-3-(quinolin-3-yl)acrylonitrile **15**

Compound **15** was prepared using above-described method from 2-cyano- methylbenzimidazole **2** (0.30 g, 1.91 mmol) and quinoline-3-carbaldehyde 12 (0.30 g, 1.91 mmol) in absolute ethanol (6 mL) to yield 0.47 g (82%) of yellow powder. m.p. 290–292 °C. ^1^H NMR (600 MHz, DMSO): δ/ppm = 13.16 (s, 1H, NH_benzimid._), 9.34 (d, *J* = 2.22 Hz, 1H, H_arom._), 8.98 (s, 1H, H_arom._), 8.55 (s, 1H, H_arom._), 8.14 (d, *J* = 7.82 Hz, 1H, H_arom._), 8.11 (d, *J* = 8.34 Hz, 1H, H_arom._), 7.91 (t, *J* = 8.33 Hz, 1H, H_arom._), 7.73 (t, *J* = 7.98 Hz, 2H, H_arom._), 7.61 (bs, 1H, H_arom._), 7.29 (bs, 2H, H_arom._); ^13^C NMR (75 MHz, DMSO): δ/ppm = 150.9, 148.5, 147.5, 142.6, 137.0, 132.2, 129.7, 129.3, 128.3, 127.3, 126.8, 124.4, 123.0, 119.9, 116.5, 112.2, 105.0; found: C, 77.15; H, 4.20; N, 18.70. Calc. for C_19_H_12_N_4_: C, 77.01; H, 4.08; N, 18.91%; MS: *m/z* = 297.1143 ([M + H]^+^).

#### 3.4.2. Benzo[g]benzo[4,5]imidazo[1,2-a][1,8]naphthyridine-6-carbonitrile **17**

Compound **17** (0.45 g, 1.35 mmol) was dissolved in sulfolane (2.5 mL), and reaction mixture was heated for 30 min at 280 °C. The cooled mixture was poured into water (10 mL), and the resulting product was filtered off and recrystallized from ethanol to obtain a yellow powder (0.39 g, 99%). m.p. 291–293 °C. ^1^H NMR (600 MHz, DMSO): δ/ppm = 9.30 (d, *J* = 7.62 Hz, 1H, H_arom._), 9.16 (s, 1H, H_arom._), 8.88 (s, 1H, H_arom._), 8.31 (d, *J* = 8.43 Hz, 1H, H_arom._), 8.26 (d, *J* = 8.02 Hz, 1H, H_arom._), 8.04 (t, *J* = 7.12 Hz, 1H, H_arom._), 8.01 (d, *J* = 8.04 Hz, 1H, H_arom._), 7.74 (t, *J* = 7.43 Hz, 1H, H_arom._), 7.64 (t, *J* = 7.51 Hz, 1H, H_arom._), 7.60 (t, *J* = 7.42 Hz, 1H, H_arom._); ^13^C NMR (75 MHz, DMSO): δ/ppm = 147.6, 143.8, 141.4, 140.8, 133.8, 131.2, 129.8, 128.4, 127.2, 125.8, 125.8, 125.0, 120.4, 117.3, 116.8, 115.5, 102.8; found: C, 77.35; H, 3.56; N, 19.08. Calc. for C_19_H_10_N_4_: C, 77.54; H, 3.42; N, 19.04%; MS: *m/z* = 3295.0958([M + H]^+^).

#### 3.4.3. Benzo[g]benzo[4,5]imidazo[1,2-a][1,5]naphthyridine-7-carbonitrile **18**

An ethanolic solution (400 mL) of compound **15** (0.35 g, 1.18 mmol) was irradiated at room temperature with 400 W high-pressure mercury lamp and using a Pyrex filter for 4 h. The solution was concentrated under reduced pressure, and resulting product was filtered off to give 0.42 g (72%) of yellow powder. m.p. 243–246 °C. ^1^H NMR (600 MHz, DMSO): δ/ppm = 9.40 (d, *J* = 2.71 Hz, 1H, H_arom._), 8.96 (d, *J* = 3.12 Hz 1H, H_arom._), 8.80–8.77 (m, 1H, H_arom._), 8.27 (d, *J* = 8.32 Hz, 1H, H_arom._), 8.18 (dd, *J_1_* = 8.41 Hz, *J_2_* = 4.14 Hz, 1H, H_arom._), 8.08 (d, *J* = 7.93 Hz, 1H, H_arom._), 8.04 (t, *J* = 7.52 Hz, 1H, H_arom._), 7.80 (t, *J* = 7.40 Hz, 1H, H_arom._), 7.66 (t, *J* = 7.62 Hz, 1H, H_arom._), 7.48 (t, *J* = 7.43 Hz, 1H, H_arom._); ^13^C NMR (150 MHz, DMSO): δ/ppm = 150.7, 148.5, 146.8, 144.6, 138.6, 138.0, 132.5, 131.8, 129.6, 126.3, 125.9, 124.1, 121.8, 120.6, 116.4, 115.9, 115.2, 115.1, 101.9; found: C, 77.68; H, 3.50; N, 18.82. Calc. for C_19_H_10_N_4_: C, 77.54; H, 3.42; N, 19.04%; MS: *m/z* = 295.0988 ([M + H]^+^).

#### 3.4.4. General Method for the Synthesis of Compounds **23** and **27**

A 2 N solution of sulfuric acid and compounds **17** and **18** was refluxed for 24 h. Cooled reaction mixture was poured into ice, and resulting product was filtered off.

##### Benzo[g]benzo[4,5]imidazo[1,2-a][1,8]naphthyridine-6-carboxylic acid **23**

Compound **23** was prepared using above-described method from **17** (0.42 g, 1.43 mmol) and 2 N aqueaus solution of sulfuric acid (4.09 mL) to yield 0.43 g (97%) of yellow powder. m.p. 287–290 °C. ^1^H NMR (300 MHz, DMSO): δ/ppm = 9.49 (s, 1H, H_arom._), 9.46 (bs, 1H, H_arom._), 9.16 (s, 1H, H_arom._), 8.34 (d, *J* = 8.62 Hz, 1H, H_arom._), 8.27 (d, *J* = 8.22 Hz, 1H, H_arom._), 8.11–8.04 (m, 2H, H_arom._), 7.79 (t, *J* = 7.43 Hz, 1H, H_arom._), 7.78–7.73 (m, 2H, H_arom._), 4.62 (bs, 1H_COOH_); found: C, 72.90; H, 3.65; N, 13.36. Calc. for C_19_H_11_N_3_O_2_: C, 72.84; H, 3.54; N, 13.51%.

##### Benzo[g]benzo[4,5]imidazo[1,2-a][1,5]naphthyridine-7-carboxylic acid **27**

Compound **27** was prepared using above-described method from **18** (0.40 g, 1.36 mmol) and 2 N aqueaus solution of sulfuric acid (3.90 mL) to yield 0.28 g (68%) of yellow powder. m.p. 290–291 °C. ^1^H NMR (400 MHz, DMSO): δ/ppm = 9.79 (s, 1H, H_arom._), 9.43 (s, 1H, H_arom._), 8.81 (d, *J* = 7.92 Hz, 1H, H_arom._), 8.39 (d, *J* = 7.63 Hz, 1H, H_arom._), 8.34 (d, *J* = 8.61 Hz, 1H, H_arom._), 8.22 (d, *J* = 8.11 Hz, 1H, H_arom._), 8.17 (t, *J* = 8.32 Hz, 1H, H_arom._), 7.93–7.86 (m, 2H, H_arom._), 7.68 (t, *J* = 8.52 Hz, 1H, H_arom._), 7.12 (bs, 1H, H_COOH_); ^13^C NMR (101 MHz, DMSO): δ/ppm = 164.3, 152.0, 148.8, 145.0, 139.6, 138.4, 135.1, 134.0, 130.3, 123.0, 129.3, 127.1, 124.9, 124.4, 117.5, 117.3, 117.0, 116.8, 116.5; found: C, 72.77; H, 3.46; N, 13.70. Calc. for C_19_H_11_N_3_O_2_: C, 72.84; H, 3.54; N, 13.51%.

#### 3.4.5. General Method for the Synthesis of Compounds **24** and **28**

A mixture of corresponding carboxylic acids **23** and **27** and thionyl chloride in absolute toluene was refluxed for 22 h.

Toluene and excess thionyl chloride were removed under reduced pressure. The crude product was washed 3 times with absolute toluene to obtain the powdered product.

##### Benzo[g]benzo[4,5]imidazo[1,2-a][1,8]naphthyridine-6-carbonyl Chloride **24**

Compound **24** was prepared using above-described method, from **23** (0.24 g, 0.80 mmol), absolute toluene (15 mL), and 0.58 mL (8.00 mmol) thionyl chloride to yield 0.19 g (72%) of yellow powder.

##### Benzo[g]benzo[4,5]imidazo[1,2-*a*][1,5]naphthyridine-7-carbonyl Chloride **28**

Compound **28** was prepared using above-described method, from **27** (0.20 g, 0.67 mmol), absolute toluene (100 mL), and 0.48 mL (6.070 mmol) thionyl chloride to yield 0.17 g (77%) of yellow powder.

#### 3.4.6. General Method for the Synthesis of Compounds **25** and **26**

A mixture of carbonyl chloride **24** and an excess of the corresponding amine in dry dichloromethane was stirred at room temperature for 2 h. The mixture was washed with 10 mL of 20% Na_2_CO_3_ and 10 mL water. After drying over MgSO_4_, the organic layer was concentrated at reduced pressure, and the resulting product was separated by column chromatography on SiO_2_ using dichloromethane/methanol as eluent.

##### *N*-(3-*N,N*-(dimethylamino)propyl)benzo[*g*]benzo[4,5]imidazo[1,2-*a*][1,8]-naphthyridine-6-carboxamide **25**

Compound **25** was prepared using above-described method from **24** (0.19 g, 0.57 mmol), dry dichloromethane (20 mL), and 0.19 mL (1.73 mmol) *N*,*N*-dimethyl- aminopropyl-1-amine to obtain 0.04 g (20%) of light orange powder. m.p. 208–212 °C. ^1^H NMR (400 MHz, DMSO): δ/ppm = 10.34 (t, *J* = 5.81 Hz, 1H, H_amide_), 9.41 (d, *J* = 7.13 Hz, 1H, H_arom._), 9.32 (s, 1H, H_arom._), 8.81 (s, 1H, H_arom._), 8.32 (d, *J* = 8.53 Hz, 1H, H_arom._), 8.22 (d, *J* = 7.80 Hz, 1H, H_arom._), 8.04–7.99 (m, 2H, H_arom._), 7.74 (t, *J* = 7.12 Hz, 1H, H_arom._), 7.68–7.61 (m, 2H, H_arom._), 3.59 (q, 2H, CH_2_), 2.88 (t, *J* = 7.40 Hz, 2H, CH_2_), 2.57 (bs, 6H, CH_3_), 2.00– 1.93 (m, 2H, CH_2_); ^13^C NMR (101 MHz, DMSO): δ/ppm = 162.3, 147.3, 146.9, 145.9, 142.7, 141.4, 134.9, 133.2, 130.8, 129.5, 128.5, 127.0, 126.1, 125.8, 124.9, 121.4, 119.8, 117.6, 117.3, 55.8, 43.9 (2C), 37.3, 25.8; found: C, 72.70; H, 5.90; N, 17.40. Calc. for C_24_H_23_N_5_O: C, 72.52; H, 5.83; N, 17.62%; MS: *m/z* = 398.1974 ([M + H]^+^).

##### Benzo[g]benzo[4,5]imidazo[1,2-a][1,8]naphthyridin-6-yl(piperidin-1-yl)metha-none **26**

Compound **26** was prepared using above-described method from **24** (0.21 g, 0.65 mmol), dry dichloromethane (20 mL), and 0.19 mL (1.93 mmol) piperidine to obtain 0.02 g (9%) of light orange powder. m.p. 258–261 °C. ^1^H NMR (400 MHz, DMSO): δ/ppm = 9.42 (d, *J* = 7.61 Hz, 1H, H_arom._), 9.11 (s, 1H, H_arom._), 8.36 (d, *J* = 8.53 Hz, 1H, H_arom._), 8.24 (d, *J* = 7.92 Hz, 1H, H_arom._), 8.09 (s, 1H, H_arom._), 8.02–7.98 (m, 2H, H_arom._), 7.74 (t, *J* = 7.11 Hz, 1H, H_arom._), 7.64 (t, *J* = 7.62 Hz, 1H, H_arom._), 7.59 (t, *J* = 7.58 Hz 1H, H_arom._), 3.74 (bs, 4H, CH_2_), 1.67 (s, 4H, CH_2_), 1.50 (bs, 2H, CH_2_); ^13^C NMR (101 MHz, DMSO): δ/ppm = 164.0, 146.6, 146.3, 145.8, 144.3, 138.7, 132.4, 131.3, 129.0, 128.4, 128.1, 128.1, 126.8, 126.0, 125.4, 124.5, 120.3, 117.9, 117.3, 48.1, 42.6, 26.5, 25.8, 24.5; found: C, 75.50; H, 5.40; N, 14.84. Calc. for C_24_H_20_N_4_O: C, 75.77; H, 5.30; N, 14.73%; MS: *m/z* = 381.1719 ([M + H]^+^).

#### 3.4.7. General Method for the Synthesis of Compounds **29** and **30**

A mixture of carboxylic acid **27**, thionyl chloride, and DMF in absolute toluene was stirred at room temperature for 10 min; after that, the appropriate amine was added and heated at reflux. Toluene and excess thionyl chloride were removed under reduced pressure. The crude product was washed 3 times with toluene to obtain powdered product. The resulting product was separated by column chromatography on SiO_2_ using dichloromethane/methanol as eluent.

##### N-isobutylbenzo[g]benzo[4,5]imidazo[1,2-a][1,5]naphthyridine-7-carboxamide **29**

Compound **29** was prepared using above-described method from **27** (0.07 g, 0.23 mmol), thionyl chloride (0.03 mL, 0.46 mmol), absolute toluene (20 mL), 0.03 mL of DMF, and 0.09 mL (1.06 mmol) of isobutyamine to obtain 0.03 g (40%) of yellow powder. m.p. 166–175 °C. ^1^H NMR (400 MHz, DMSO): δ/ppm = 10.45 (t, *J* = 5.80 Hz, 1H, H_NH_), 9.59 (s, 1H, H_arom._), 8.93 (s, 1H, H_arom._), 8.77 (d, *J* = 8.42 Hz, 1H, H_arom._), 8.28 (dd, *J_1_* = 8.42, *J_2_* = 0.91 Hz, 1H, H_arom._), 8.19 (d, *J* = 8.54 Hz, 1H, H_arom._), 8.06–7.99 (m, 2H, H_arom._), 7.78 (t, *J* = 8.36 Hz, 1H, H_arom._), 7.66 (t, *J* = 7.22 Hz, 1H, H_arom._), 7.47 (t, *J* = 8.48 Hz, 1H, H_arom._), 3.39 (t, *J* = 6.14 Hz, 2H, H_CH2_), 2.01–1.99 (m, 1H, H_CH_), 1.06 (d, *J* = 6.78 Hz, 6H, H_CH3_); ^13^C NMR (101 MHz, DMSO) δ/ppm = 162.0, 152.3, 148.5, 148.2, 144.0, 137.7, 132.9, 132.2, 131.6, 130.1, 126.7, 126.0, 124.4, 122.0, 121.5, 120.5, 116.9, 116.3, 116.3, 47.1, 29.1, 20.6.

##### *N,N*-dimethylbenzo[g]benzo[4,5]imidazo[1,2-a][1,5]naphthyridine-7-carbox-amide **30**

Compound **30** was prepared using above-described method from **27** (0.20 g, 0.64 mmol), thionyl chloride (0.10 mL, 1.28 mmol), absolute toluene (50 mL), 0.10 mL of DMF, and 0.11 mL (3.19 mmol) of methylamine to obtain 0.11 g (48%) of light yellow powder. m.p. 160–165 °C. ^1^H NMR (400 MHz, DMSO): δ/ppm = 9.46 (s, 1H, H_arom._), 8.83 (d, *J* = 8.43 Hz, 1H, H_arom._), 8.29 (d, *J* = 8.42 Hz, 1H, H_arom._), 8.23 (s, 1H, H_arom._), 8.22 (d, *J* = 7.20 Hz, 1H, H_arom._), 8.06 (d, *J* = 7.89 Hz, 1H, H_arom._), 8.00 (t, *J* = 8.31 Hz, 1H, H_arom._), 7.81 (t, *J* = 8.29 Hz, 1H, H_arom._), 7.63 (t, *J* = 8.12 Hz, 1H, H_arom._), 7.47 (t, *J* = 8.42 Hz, 1H, H_arom._), 3.16 (s, 3H, CH_3_), 2.99 (s, 3H, CH_3_); ^13^C NMR (101 MHz, DMSO): δ/ppm = 165.7, 151.5, 148.1, 147.2, 145.4, 136.9, 132.1, 131.6, 130.1, 128.1, 127.5, 126.1, 126.0, 124.2, 121.7, 121.0, 117.3, 116.6, 116.2, 38.6, 35.0; found: C, 74.30; H, 4.58; N, 16.62. Calc. for C_21_H_16_N_4_O: C, 74.10; H, 4.74; N, 16.46%; MS: *m/z* = 341.1408 ([M + H]^+^).

### 3.5. Antiproliferative Activity

#### 3.5.1. Cell Culture and Reference Compounds

Human cancer cell lines used in this manuscript, namely, Capan-1, HCT-116, NCI-H460, LN-229, LS513, HL-60, K-562, and Z-138, were acquired from the American Type Culture Collection (ATCC, Manassas, VA, USA), while the DND-41 cell line was purchased from the Deutsche Sammlung von Mikroorganismen und Zellkulturen (DSMZ Leibniz-Institut, Leibniz, Germany). Culture media were purchased from Gibco Life Technologies, USA, and supplemented with 10% fetal bovine serum (HyClone, GE Healthcare Life Sciences, Marlborough, MA, USA). Etoposide and doxorubicin, which were used as reference inhibitors, were purchased from Selleckchem (Munich, Germany). Stock solutions were prepared in DMSO.

#### 3.5.2. Proliferation Assays

Adherent cell lines Hap-1 and Capan-1 cells were seeded at a density of 500 cells per well, whereas HCT-116 and NCI-H460 were seeded at 1500 cells per well in 384-well tissue culture plates (Greiner, Kremsmünster, Austria). After overnight incubation, cells were treated with seven different concentrations of the test compounds, ranging from 100 to 0.006µM. Suspension cell lines HL-60, K-562, Z-138, and MM.1S were seeded at 2500 cells per well, and for the DND-41 cell line, 5500 cells were seeded in 384-well culture plates containing the test compounds at the same concentration points. Cells were incubated for 72 h with compounds and were then analyzed using the CellTiter 96^®^ AQueous One Solution Cell Proliferation Assay (MTS) reagent (Promega, Fitchburg, WI, USA) according to the manufacturer’s instructions. The absorbance of the samples was measured at 490 nm using a SpectraMax Plus 384 (Molecular Devices, San Jose, CA, USA), and OD values were used to calculate the 50% inhibitory concentration (IC_50_). Compounds were tested in at least two independent experiments.

#### 3.5.3. High Content Imaging Analysis of Cell Cycle Distribution

Capan-1 and NCI-H460 cells were seeded at 5000 cells per well in 96 well-clear flat bottom tissue culture plates. After overnight incubation, the cells were treated with the test compounds at different concentrations for 24 h. Cells were then fixed with 4% PFA in PBS for 10 min and washed, and the nuclei were stained by adding a solution of 300 nM 4′,6-diamidino-2-phenylindole (DAPI, Molecular Probes, Eugene, OR, USA). The plates were imaged on a CX5 High Content Screening device (ThermoFisher Scientific, Waltham, MA, USA) using the Cell Cycle Analysis bio-application. A minimum of 500 cells was imaged for each well.

#### 3.5.4. Caspase-3/7 Activity

Caspase activity was assayed by measuring light intensity using Caspase-Glo^®^ 3/7 (Promega Corporation, Fitchburg, WI, USA) according to the manufacturer’s protocol. Briefly, Capan-1 and NCI-H460 cells were seeded at 5000 cells per well in white 96-well microplates and treated with the test compounds at different concentrations for 24 h. Caspase-Glo reagent was added and incubated at room temperature for 30 min. The caspase activity was then measured using the GloMax^®^-96 (Promega Corporation, Fitchburg, WI, USA) microplate luminometer, recording the mean of relative light units (RLU) per well. Consequently, the nuclei were stained by adding Hoechst 33342 (ThermoFisher Scientific, Waltham, MA, USA) at a final concentration of 1 µg/mL, and after another 30 min incubation period, a CX5 High Content Screening device (ThermoFisher Scientific) was employed to measure the total amount of cells per well to facilitate the calculation of the apoptotic rate.

### 3.6. Binding with Nucleic Acids

The UV/Vis spectra were recorded on a Varian Cary 100 Bio spectrophotometer (Agilent, Santa Clara, CA, USA), CD spectra on a JASCO J815 spectrophotometer (ABL&E Handels GmbH, Wien, Austria), and fluorescence spectra on a Varian Cary Eclipse spectrophotometer (Agilent, Santa Clara, CA, USA) at 25 °C using appropriate 1 cm path quartz cuvettes. Calf thymus DNA, ctDNA, and rArU were purchased from Sigma-Aldrich (St. Louis, MI, USA) and dissolved in Na-cacodylate buffer, *I* = 0.05 mol dm^−3^, pH = 7.0. The ctDNA was additionally sonicated and filtered through a 0.45 mm filter [36]. DNA and RNA concentration was determined spectroscopically as the concentration of phosphates [37].

Spectrophotometric titrations were performed at pH = 7.0 (*I* = 0.05 mol dm^−3^, sodium cacodylate buffer) by adding portions of polynucleotide solution into the solution of the studied compound for fluorimetric experiments, and CD experiments were performed by adding portions of the compound stock solution into the solution of a polynucleotide. In fluorimetric experiments, an excitation wavelength of λ_exc_ ≥ 300 nm was used to avoid the inner filter effect caused by increasing absorbance of the polynucleotide. Emission was collected in the range λ_em_ = 450–750 nm. Values for *K*_a_ were obtained by processing titration data using the Scatchard equation (Table 2); most of them have satisfactory correlation coefficients (>0.99). Thermal melting curves for DNA and their complexes with studied compounds were determined as previously described by following the absorption change at 260 nm as a function of temperature. The absorbance of the ligands was subtracted from every curve, and the absorbance scale was normalized. *T*_m_ values are the midpoints of the transition curves determined from the maximum of the first derivative and checked graphically by the tangent method. The Δ*T*_m_ values were calculated by subtracting the *T*_m_ of the free nucleic acid from the *T*_m_ of the complex. Every Δ*T*_m_ value reported here was the average of at least two measurements. The error in Δ*T*_m_ is ±0.5 °C.

## 4. Conclusions

This manuscript details the synthesis, structural characterization, assessment of anti-proliferative activity, and examination of DNA/RNA binding properties in pentacyclic benzimidazole derivatives containing amino or amido side chains.

The amino side chains were placed at positions 7 or 11 of the pentacyclic skeleton, whereas amido side chains were positioned at positions 6 or 7, with variations in the positioning of the N atom on the quinoline nuclei. The synthesized compounds underwent in vitro evaluation for their antiproliferative activity, and for the most promising compounds, investigations were carried out to elucidate their binding modes with DNA/RNA, aiming to uncover their mechanisms of biological action. Synthesis of the target compounds employed conventional organic synthesis techniques alongside environmentally friendly methods such as photochemical cyclization or microwave-assisted amination.

Only compounds soluble in DMSO at concentrations 1 × 10^−2^ M were subjected to in vitro testing for their antiproliferative effects on various human cancer cell lines. The most promising compound identified was the *N,N*-dimethylaminopropyl-substituted derivative **6**, carrying an amino side chain at position 7 of the pentacyclic skeleton. This compound exhibited noteworthy inhibition of proliferation across multiple cancer cell lines, with inhibitory concentrations ranging from 0.3 to 1.8 µM. Additionally, structural analogue **19**, featuring an amino side chain at position 11 of the pentacyclic skeleton, also demonstrated potent activity, particularly against K-562 and Z-138 cancer cells, with inhibitory concentrations ranging from 0.4 to 5.9 µM. Moreover, both the 7- and 11-piperazinyl-substituted pentacyclic derivatives, compounds **9** and **22**, respectively, showcased robust and extensive antitumoral activity. In summary, the obtained results suggest that the aliphatic amino chain positioned at position 7 enhanced the observed antiproliferative activities in comparison to the analogues substituted at position 11, while the compounds substituted with a cyclic amino side chain at position 11 showed slightly improved antiproliferative activity compared to analogues substituted at the position 7 of the pentacyclic skeleton.

Thermal melting experiments, fluorescence, and circular dichroism spectroscopy collectively indicate a dual binding mode—comprising intercalation and the binding of aggregated compounds along the polynucleotide backbone—for the examined drugs concerning both DNA and RNA. The strength of these interactions with DNA/RNA was significantly influenced by the positioning of the amino substituent on the pentacyclic ring. Specifically, derivatives **19** and **22**, featuring an amino substituent (piperazine or *N,N*-dimethylaminopropylamino) at C-11, demonstrated enhanced binding affinities and thermal stabilization effects toward double-stranded DNA (ds-DNA) and double-stranded RNA (ds-RNA) in comparison to derivatives **6** and **9** with substituents at C-7 (as outlined in Table 2). The considerable binding affinities observed for compounds **19** and **22** suggest that direct interaction with DNA/RNA might represent their biological targets. Additional experiments to evaluate the effects of the most promising pentacyclic benzimidazole derivatives **6** and **9** on both cancer cell cycle regulation and caspase-3/7 activity yielded results paralleling those of the included reference drug doxorubicin, also confirming a similar mode of action by direct interaction with nucleic acids.

## Data Availability

Additional data are available on request.

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
