# Peer review of "Synthesis and Biological Evaluation of Novel Amino and Amido Substituted Pentacyclic Benzimidazole Derivatives as Antiproliferative Agents"

_ijms, 2024, doi:10.3390/ijms25042288_

Round 1
Reviewer 1 Report (Previous Reviewer 2)
Comments and Suggestions for Authors
See the attachment

1. In general, the manuscript appears written with low care and needs many corrections; several parts of the manuscript are not easily understandable because of a not appropriate English form.
Author Response
I carefully revised the new version of the manuscript by Perin et al and the Authors’ responses to referees’ concerns. In this Reviewer’s opinion, the paper has been improved; however some points have to be addressed before the acceptance for publication.
- In general, the manuscript appears written with low care and needs many corrections; several parts of the manuscript are not easily understandable because of a not appropriate English form; the reviewer suggests to carefully revise the English form of the whole manuscript.
Answer: Thank you for this comment. We have done extenisve revision of the English language throught the whole manuscript.
ONLY PARTIALLY DONE.
Some parts of the manuscript have to be revised.
As few examples:
- Page 1, first two lines, Introduction. “Nowadays, nitrogen heterocycles play an essential role as fundamental structures in numerous synthetic and natural pharmacological compounds”. This sentence has to be substituted with “Nowadays, nitrogen heterocycles are the essential core in numerous synthetic and natural pharmacologically active compounds”.
- Page 1, lines 2-3, Introduction. “They are frequently utilized in the strategic development of new biologically active molecules”. This sentence has to be deleted.
- Page 1, line 4, Introduction. Substitute “characteristics” with “activities”.
- Page 1, line 9, Introduction. Substitute “which are not only vital in medicinal chemistry” with “which are exploitable in medicinal chemistry”.
- Page 1, line 11, Introduction. Substitute “engage” with “interact”.
- Page 1, line 12, Introduction. Delete “among”.
- Page 2, lines 9-10, Introduction. Change “..we have recently synthesized a series of compounds. These include tetracyclic quinoline-fused...” with “we have recently synthesized a series of compounds, including tetracyclic quinoline-fused”.
- Page 3, first two lines, Results and Discussion, Chemistry. “All targeted amino-substituted compounds at positions 6-9 and 19-22, as well as amido-substituted compounds at positions 25-26 and 29-30 in…”. Delete “at positions”.
- Page 3, line 15, Results and Discussion, Chemistry. Change “yielding moderate results” with “with yields”.
- Page 3, second line from the bottom, Results and Discussion, Chemistry. Substitute “yielding in the range” with “with”.
- Page 4, first two lines, Results and Discussion, Chemistry. “Additionally, two distinct synthetic methods were employed for preparing the targeted 6-amido substituted pentacyclic derivatives”. This sentence has to be revised being the two distinct methods employed to synthesize 6- and 7-amido substituted compounds.
- Page 5, line 7, Biological activity, Antiproliferative activity in vitro. “Substitute “demonstrating” with “reporting”.
- Page 5, line 14, Biological activity, Antiproliferative activity in vitro. Delete “Among these compounds”.
- Page 6, lines 12-14 from the bottom, Biological activity, Antiproliferative activity in vitro. “Among the amido-substituted compounds, the carboxamide derivative 25, featuring an N,N-dimethylaminopropyl side chain, demonstrated the highest activity, exhibiting an IC50 range of 2.1 – 11.9 μM”. Pay attention to IC50 values.
- Page 20, lines 12-13, conclusion. “Only soluble compounds were subjected to in vitro testing for their antiproliferative effects on various human cancer cell lines”. Soluble in which solvent? At which concentration? Please specify, also in the main text.
- Page 20, lines 12-1, conclusion. “This compound exhibited noteworthy inhibition of proliferation across multiple cancer cell lines, with inhibitory concentrations ranging from 0.3 to 0.6 μM”. Check the IC50 values.
- Page 20, lines 3-4 from the bottom, conclusion. “Additional experiments to evaluate the effects of the most promising pentacyclic benzimidazole derivatives 6 and 19 on both cancer cell cycle regulation and caspase-3/7…”. 19 has to be replaced with 9.
- Reduced the number of “notably”.
- Change targeted with target.
Answer: We have corrected everything as it was suggested point by point also done some additional revision for some parts of manuscript. We do hope that now the quality of English language will be satisfied.
Results and Discussion
- b) to make easier for the reader to follow the SAR, please insert a figure with all the formulas of final compounds with the appropriate numbering of the amino and amido substituents.
Answer: We have inserted a new figure as it was suggested.
NOT DONE. The figure 2 is not representative of the SAR. Please revise.
Answer: We have revised and corrected the Figure 2.
- g) It is not clear why the authors have synthesized compound 26 with a piperidinyl amido group and compound 30 with a dimethyl amido group instead of a piperazinyl amido one to make a rational comparison with the activity of the highly antiproliferative amino derivatives 9 and 22: please explain. Moreover, these two piperazinyl amido derivatives could have aid in making a sound SAR of the amido regioisomers 25-26 and 29-30 that is completely lacking.
Answer: We have problems with the synthesis of mentioned piperazinyl substited derivative and the synthesis failed after several attempting. Also, we got the N,N-dimethyl substituted derivative as a results of the reactivity of solvent DMF, namely alkylation could happen at the higher temperatures (aready described in the literature).
NOT DONE. Please explain in the text why compound with the piperazinyl amido group has not been synthesized.
Answer: We have explained in text by adding following sentence: “Due to the synthetic issues, the piperazinyl substituted compound was not synthesized, while compound 30 obtained via the interaction of DMF with carbonyl chloride 28 upon microwave heating.”
- h) The conclusions of the 2.2.1 paragraph are confusing: please completely rewrite.
Answer: We have rewritten 2.2.1. paragraph.
NOT DONE. The conclusions of the 2.2.1 paragraph are still confusing. Please completely rewrite.
Answer: We have rewritten this paragraph.
- Data concerning the antiproliferative activity and reported in table 1, page 6, should be associated with SEM or SD.
Answer: Table 1 is updated with SD as suggested.
Some SDs are very similar to the measured values. Please revise.
Answer: Revising” SD values is impossible; they are what they are. In order to get other values, the experiments need to be repeated which is not possible at the time.
Conclusion
The sentence “Summarizing….activities” is in contrast with what asserted in the paragraph: please completely rewrite the whole paragraph.
Answer: We have rewritten as suggested.
NOT DONE. The sentence is still in contrast with what asserted in the paragraph. Please rewrite.
Answer: We have carefully checked and rewrite this sentence to: “In summary, the obtained results suggest that the aliphatic amino chain positioned at position 7 enhanced the observed antiproliferative activities in comparison to the analogues substituted at the position 11, while the compounds substituted with cyclic amino side chain at the position 11 showed slightly improved antiproliferative activity compared to analogues substituted at the position 7 of pentacyclic skeleton.
Experimental
- a) The 1H-NMR of compounds 4, 6, 20 and 30 has to be checked.
Answer: We have checked NMR dana and corrected.
ONLY PARTIALLY DONE. The 1H-NMR of compound 20 has to be checked.
Answer: We have checked and corrected the 1H NMR spectra of compounds 20.
Materials and Methods
Antiproliferative activity, Proliferation assays, pages 16-17.
Why the density at which the different cell lines are seeded is so variable (500-1500 cells per well)? This can lead to very different results. The Authors are invited to deeply explain the conditions used for this assay.
Answer: Different cell lines have distinct growth characteristics, doubling times, and optimal densities for proliferation. Some cell lines may require higher or lower seeding densities to ensure they reach an optimal confluence during the assay period. In this case, Hap-1 and Capan-1 cells have higher growth rates when compared to HCT-116 and NCI-H460 cell lines, and hence are seeded at a lower density to prevent overgrowth of the well plates and/or nutrient depletion.
We agree with the authors, but is pivotal to modify the method section. Please indicate the specific density use for the single cell line.
Answer: We added the requested informations.

Reviewer 2 Report (Previous Reviewer 3)
Comments and Suggestions for Authors
The authors have modified the manuscript significantly and answered all questions. I don’t have any further concerns and I recommend this manuscript for publication.
Author Response
The authors have modified the manuscript significantly and answered all questions. I don’t have any further concerns and I recommend this manuscript for publication.
Answer: We would like to thank the reviewer.

Round 2
Reviewer 1 Report (Previous Reviewer 2)
Comments and Suggestions for Authors
see the attached file

This manuscript is a resubmission of an earlier submission. The following is a list of the peer review reports and author responses from that submission.
Round 1
Reviewer 1 Report
Comments and Suggestions for Authors
The article by Nataša Perin et al. refers to Synthesis and biological evaluation of novel amino and amido substituted pentacyclic benzimidazole derivatives.
In this manuscript, the authors present the synthesis and biological evaluation of a series of pentacyclic benzimidazole derivatives with amino or amido side chains. As well as their antiproliferative activity in vitro on heterogeneous panel of cancer cell lines. A total of 16 compounds were evaluated. In addition, they present their direct binding to nucleic acids to elucidate their possible mechanism of biological action. All compounds were prepared by conventional organic synthesis reactions, by photochemical reactions and also by microwave-assisted reactions.
On the other hand, even though the outcome is not so exciting, this work is well designed and properly oriented so I would like to appreciate the effort that has been made.
However, it is recommended to accept the manuscript for publication after major revision, to improve the quality of the work.
Therefore, it is recommended:
1.- It is highly recommended to number those pentacyclic structures that give rise to the products that are then evaluated in vitro, in order to be able to adequately follow the results and conclusions presented in the manuscript.
2.- Secondly, since to obtain the final products to be evaluated the cyclization of the acyclic hydroxyl substituted acrylonitrile derivative 3 and the acyclic acrylonitrile 13 are essential. It is recommended to include the reaction mechanism that allows the formation of pentacyclic benzimidazole derivatives in the presence of t-KOBu in DMF, as well as its formation in the presence of hv/EtOH and include it in the manuscript with its respective comments.
3.- Finally, the authors report in the experimental part of the manuscript approximately seven compounds that have melting point values above 300 oC. As is well known, most organic compounds have melting points below or close to 250 oC. What do you think is the reason why these high melting point values are obtained? If possible, include it in the manuscript where it is most appropriate to report it.

Reviewer 2 Report
Comments and Suggestions for Authors
The paper by Perin et al. describes the synthesis of a series of novel pentacyclic benzimidazole derivatives decorated with either amino or amido side chains. All compounds were prepared by conventional reactions of organic synthesis as well as by environmentally friendly methods, that is photochemical and microwave assisted reactions. All the newly synthesized compounds were tested for their antiproliferative activity in vitro on several human cancer cell lines, and compound 6 showed a remarkable potency in the sub-micromolar range in most of the tested cancer cell lines. SARs evidenced that the biological activity was strongly influenced by the position and type of side chain at the pentacyclic skeleton. By means of fluorescence, circular dichroism spectroscopy and thermal melting assays, the authors suggested a mixed binding mode (intercalation and binding of aggregated compounds along the polynucleotide backbone) to DNA and RNA for these pentacyclic benzimidazoles.
Although the presented pentacyclic benzimidazole chemical scaffold is unique and the obtained results may fall in the scope of the journal, in this reviewer opinion, in its present form the paper is not suitable for publication and needs major revisions.
Major points:
1. In general, the manuscript appears written with low care and needs many corrections; several parts of the manuscript are not easily understandable because of a not appropriate English form; the reviewer suggests to carefully revise the English form of the whole manuscript.
2. In the Introduction section several references are not updated: please revise.
3. SAR discussion is almost poor and confused:
a) numbers of compounds are never cited through the text: please insert.
b) to make easier for the reader to follow the SAR, please insert a figure with all the formulas of final compounds with the appropriate numbering of the amino and amido substituents.
c) some sentences have no meaning: please rewrite.
d) some numbers of compounds are wrong: please check.
e) IC50s of compounds are not properly reported: for example, compound 6 (IC50 values 0.3 – 1.8 mM) and not (IC50 values 0.3 – 0.6 mM);
f) Selectivity towards certain cancer cell lines is cited, but, actually, it’s only a higher antiproliferative activity towards certain cancer cell lines; please rewrite.
g) It is not clear why the authors have synthesized compound 26 with a piperidinyl amido group and compound 30 with a dimethyl amido group instead of a piperazinyl amido one to make a rational comparison with the activity of the highly antiproliferative amino derivatives 9 and 22: please explain. Moreover, these two piperazinyl amido derivatives could have aid in making a sound SAR of the amido regioisomers 25-26 and 29-30 that is completely lacking.
h) The conclusions of the 2.2.1 paragraph are confusing: please completely rewrite.
4. Data concerning the antiproliferative activity and reported in table 1, page 6, should be associated with SEM or SD.
5. Biological activity, binding with nucleic acids, pages 7-9. To validate results concerning the ability of compounds to interact with nucleic acid, the Authors should add a positive control in the assay. For example: Rhodamine B (minor groove binder) and Methyl Green (major groove binder). See references doi: 10.1016/s.dyepig.2013.05.28 and 10.1007/s00418-014-1215-0.
6. Conclusion:
a) The paragraph “Acyclic….chlorides” has no meaning: please delete.
b) The sentence “Summarizing….activities” is in contrast with what asserted in the paragraph: please completely rewrite the whole paragraph.
c) Authors assert: “Derivatives 19 and 22 with an amino substituent (piperazine or N,N-dimethylaminopropylamino) at C-11 showed improved binding affinities and thermal stabilization effects towards ds-DNA and ds-RNA when compared to derivatives 6 and 9 with substituents at C-7 (Table 2). The high binding affinities obtained for 19 and 22 suggest that the direct interaction with DNA/RNA might be the biological targets.” However, compounds 6 and 9 have an antiproliferative activity higher than that exerted by 19 and 22: do the authors have an explanation for this discrepancy?
7. The experimental section of the manuscript (pages 10-16) needs to be revised in several parts. For example:
a. The 1H-NMR of compounds 4, 6, 20 and 30 have to be checked.
b. The alkylamine used for the synthesis of compounds 7 and 20 is the isobutylamine and not isopropyl.
c. 1H-NMR spectrum of compound 18 is not fully integrated.
d. The synthesis of compound 29 is described twice. Also in the scheme 2 (page 4), it is not appropriately outlined.
e. The synthesis followed by the authors to obtain compound 24 lacks.
In addition, it is strongly recommended to the Authors to upload the full 1H-NMR spectra of compounds in the Supporting Information.
8. Materials and Methods, Antiproliferative activity, Proliferation assays, pages 16-17. Why the density at which the different cell lines are seeded is so variable ((500-1500 cells per well)? This can lead to very different results. The Authors are invited to deeply explain the conditions used for this assay.
Minor:
1. Abstract: “Derivatives 6 and 14 showed significant inhibitory activity…”: compound 14 should be 9 and inhibitory activity should be antiproliferative activity.
2. Scheme 2: synthesis of compounds 29 and 30: which is the right solvent, DMF or CH2Cl2?
3. Figure 2: instead of (IC50 0.3 – 0.6 M) please insert: compound 6 (IC50 values 0.3 – 1.8 mM).
Comments on the Quality of English LanguageIn general, the manuscript appears written with low care and needs many corrections; several parts of the manuscript are not easily understandable because of a not appropriate English form; the reviewer suggests to carefully revise the English form of the whole manuscript.
Reviewer 3 Report
Comments and Suggestions for Authors
In this manuscripts, the authors reported the synthesis of several anticancer agents based on pentacyclic benzimidazole. The work included well synthesized and characterized compounds and has been well written however, it has some serious flaws that make me suggest the rejection with giving an opportunity to resubmit. The following points should be addressed:
1- Some experiments need to be conducted to investigate the proposed mechanism of actions and mechanism of cell deaths for the synthesized compounds. These experiments are
a- Flow cytometry analysis for apoptosis and cell cycle analysis.
b- Determination of pro-apoptotic and anti-apoptotic markers.
2- It seems all experiments including the antiproliferative assay and the DNA binding assay have done only one time. These experiments need to be done 3 times at least and the results should include SEM or SD.
3- The design of the target compounds is random and is not based on rationale drug design therefore, the authors need to make a good link between the chemical structures of the designed compounds and the expected anticancer activity.
some minor points:
1- The title should include this phrase at the end “as antiproliferative agents”
2- In the abstract, compound 14 is mentioned as a significant antiproliferative agent however that compound has never been tested. Please correct that.
3- on the last arrows of scheme 1 and 2,, the word amine should be written as corresponding amine.
4- In table 1, after uM, add an asterisk and define it in the caption and mention that the experiment has been done 3 times at least with writing SD.
5- the binding with nucleic acid experiments need to be done on a reference compound to compare the results and validate them with target compounds.
6- why the authors incubate the target compounds with tested cell lines for 72 h?
a-the conditions of the incubation need to be addressed (temp, humidity and CO2).
b- how many ml of MTS was added? And how long was incubated after the addition?
Comments on the Quality of English Language
Minor typos